# Vaccines and Animal Models of Nipah Virus: Current Situation and Future Prospects

**DOI:** 10.3390/vaccines13060608

**Published:** 2025-06-04

**Authors:** Chaoxiang Lv, Jiayue He, Qiqi Zhang, Tiecheng Wang

**Affiliations:** 1The Research Center for Preclinical Medicine, Southwest Medical University, Luzhou 646000, China; lvchaoxiang@swmu.edu.cn (C.L.); hejieyue@swmu.edu.cn (J.H.); zhangqq966@nenu.edu.cn (Q.Z.); 2Changchun Veterinary Research Institute, Chinese Academy of Agricultural Sciences, Changchun 130122, China; 3College of Life Sciences, Shandong Normal University, Jinan 250358, China

**Keywords:** NiV, animal models, vaccine development, zoonosis, pandemic preparedness

## Abstract

Nipah virus (NiV) is a highly pathogenic paramyxovirus characterized by zoonotic infection, high mortality, and a lack of effective treatment, posing a serious threat to global public health security. Currently, it still lacks specific treatments or approved vaccines, and is listed as a potential pandemic threat pathogen by the World Health Organization. This paper systematically reviews the core progress and challenges of NiV investigation, with a focus on the development of animal models, vaccine development strategies, treatment strategy, and bottlenecks in translational medicine. Additionally, we discuss the strengths and limitations of existing animal models, including ferrets, hamsters, mice, and non-human primates (NHPs), and assess advances in vaccine platforms such as viral vectors, subunit vaccines, and mRNA-based vaccine candidates. The paper critically reviews the challenges facing translational research, conservation correlates, and outbreak preparedness, while also providing future research directions for pandemic preparedness and public health security strategies.

## 1. Introduction

Emerging infectious diseases have become a constant concern in global public health. NiV first emerged during an outbreak in Malaysia in 1998 [1]. In recent years, outbreaks of NiV have also occurred in Bangladesh, Malaysia, and India [2,3]. NiV, a member of the Paramyxoviridae family, can infect humans and various mammals and is classified as a Biosafety Level 4 pathogen [4]. It can cause several serious diseases in humans, in particular severe respiratory disease and encephalitis [5]. NiV infection is mainly transmitted through direct contact with respiratory droplets, nasal secretions, and tissues of infected animals. It can also be spread by consuming food contaminated with the urine, feces or saliva of infected bats. It has been reported that NiV infection can be transmitted from person to person, mainly through close contact with the secretions and excretions of infected patients [4]. After humans are infected with NiV, they may show symptoms such as fever, headache, dizziness, vomiting, decreased consciousness, segmental myoclonus, reduced reflexes, hypotonia, and abnormal head-eye reflexes, with a case-fatality rate as high as 75% [6]. Histopathological examination has revealed that systemic vasculitis with extensive thrombosis and parenchymal necrosis were the predominant features, with particularly prominent involvement of the central nervous system. In affected vessels, endothelial cell damage, necrosis, and syncytial giant cell formation were observed. Characteristic viral inclusions were identified via both light and electron microscopy. Immunohistochemical (IHC) analysis confirmed that Nipah virus antigens were widely present in vascular endothelial and smooth muscle cells, while abundant viral antigen expression was also detected in various parenchymal cells—most notably in neurons [7,8]. In 2001, during the first reported outbreak of Nipah virus in Siliguri, West Bengal, India, 68% of 66 infected cases resulted in death [9]. Subsequently, in the 2018 outbreak in Kerala, the case fatality rate (CFR) reached 91% among 23 patients, and in 2021, a single case in Kerala had a 100% CFR [5,10]. In 2023, two out of six confirmed cases in Kerala died, resulting in a CFR of approximately 33%. Repeated outbreaks and consistently high case fatality rates over the years continue to highlight the severe threat posed by the Nipah virus to human health, making it a critical and non-negotiable hazard in public health security. Because of its high pathogenicity and transmission risk, NiV has been listed by the WHO as one of the top 10 infectious diseases of priority concern [11]. However, to date, no licensed vaccine or therapeutic drug is available [12]. Therefore, the development of a safe and effective NiV vaccine is an urgent priority.

Current NiV vaccine development predominantly focuses on the viral G and F proteins as antigenic targets [13,14]. Multiple vaccine platforms, including viral vectors, virus-like particles, nucleic acid formulations, and nanoparticle-based systems, are under active investigation to develop safe and effective countermeasures against NiV infection [15]. These efforts hold significant promise for establishing preventive strategies against this emerging zoonotic pathogen. In preclinical research, animal models serve as critical tools for both immunogenicity assessment and infection pathophysiology studies. Various studies have shown that some candidate vaccines can provide complete protection in preclinical studies using hamster [16,17], cat [18], ferret [19], non-human primate [20,21] and pig [22] models by using one or both of NiV G and F proteins as antigens. For example, candidate vaccines using vesicular stomatitis virus vectors have been shown to have protective effects in hamsters, ferrets, and African green monkeys [12]. In these animal models, the titer of neutralizing antibodies induced by the vaccine usually ranges from 160 to 16,000 [23]. After challenge, few or no clinical symptoms of the disease were observed in vaccinated animals. Depending on the route and dose of vaccination, the protection rate can reach 100%. Mice can also produce immune responses, but they are not susceptible to NiV infection, so protection cannot be evaluated [13,23]. These experimental systems enable researchers to systematically investigate early infection dynamics while rigorously testing vaccine candidates’ protective capacity. Through controlled challenge studies, scientists can delineate immune correlates of protection, assess viral clearance mechanisms, and validate vaccine-mediated prevention of clinical disease manifestations. The complementary strengths of these models collectively advance our understanding of NiV pathogenesis and accelerate vaccine development efforts.

This paper aims to provide a comprehensive review of the current status of research on animal vaccines and animal models for NiV, detailing the progress in the development of different types of vaccines, such as subunit vaccines, recombinant live vector vaccines, and nucleic acid vaccines, as well as their immunological effects, strengths, and challenges. At the same time, the research on NiV vaccines and animal models will be taken as the core focus, with an in-depth analysis of the challenges faced by NiV vaccines during development and application. A systematic outlook on the future direction of research will be provided to offer a theoretical basis for in-depth studies on animal vaccines against NiV and serve as a reference for the future research and development of NiV vaccines.

## 2. Biological Characteristics

NiV belongs to the family Paramyxoviridae, subfamily Paramyxovirinae, and genus Henipavirus. The genus Henipavirus includes Hendra virus (HeV), Cedar virus, Kumasi virus, Mojiang virus, and Langya henipavirus, in addition to NiV [24,25] (Figure 1). NiV has only one serotype, but genetic variation has led to the evolution of two lineages of the virus during transmission, namely, the Bangladesh lineage (NiV Bangladesh, NiV-B) and the Malaysian lineage (NiV Malaysia, NiV-M), which have nucleotide similarities of up to 91.8% [26].

## 3. The Structure of NiV

NiV is a vesicular virus with an unsegmented negative-stranded RNA genome. The virus particles are polymorphic, including spherical and filamentous forms, and range in size from 180 to 1900 nm, with an average size of 500 nm [27]. Like all other paramyxoviruses, the NiV genome contains six genes corresponding to the nucleocapsid (N), phosphoprotein (P), matrix protein (M), fusion protein (F), glycoprotein (G), and large protein (L) [28]. The two lineages of NiV are roughly the same in terms of structure and function, but they have different genome sizes. The NiV-MY genome contains 18,246 nucleotides (nts), whereas the NiV-BD genome has 18,252 nts [29]. The N, P, and L proteins play a role in viral replication [30]. The accessory proteins V/W/C are encoded through the variable open reading frame and RNA editing of the P gene, where the C protein is translated by the downstream start codon of the P mRNA. Upon insertion of a guanine at the RNA editing site, the RNA of the P gene is edited to generate the V and W genes, and the C gene is encoded in a different reading frame that regulates viral RNA synthesis. The V and W proteins are thought to suppress the innate immune response by directly blocking interferon α (IFNα) signaling [31]. The M protein has functional roles in viral growth and assembly, but viral entry is mediated by two glycoproteins, F and G [32]. Amino acid similarity between NiV and HeV ranges from 68% to 92%, with 88% and 83% similarity between the F and G proteins of the two, respectively [33]. Thus, NiV and HeV exhibit significant or complete cross-reactivity [34]. The structure and genomic arrangement of NiV are shown in Figure 2.

## 4. Animal Models

The commonly used animal models for NiV evaluation include Golden Syrian hamsters, ferrets, African green monkey models, the Cavia porcellus, cats, swine, and mice, among others. Each animal model has its own advantages and limitations (Table 1).

### 4.1. Golden Syrian Hamster Model

Syrian hamsters are well-established animal models for NiV pathogenesis studies. Dose-dependent mortality analyses demonstrate distinct route-specific lethality: the median lethal dose (LD_50_) via intraperitoneal (IP) injection ranges from 1 to 10^4^ plaque-forming units (PFU), whereas intranasal (IN) inoculation requires 10^1^–10^6^ PFU to induce fatal outcomes. IP-infected hamsters exhibit accelerated disease progression, developing characteristic neurological symptoms (tremors and limb paralysis) at 5–9 days post-infection (dpi), with mortality occurring within 24 h of post-symptom onset [35]. The majority of gophers inoculated via the IN route showed progressive deterioration of limb imbalance, paralysis, lethargy, muscle twitching, and respiratory distress in the final stages, and most of the infected gophers died within 9–15 days [36]. At the end stage of the infection, pathological lesions were most severe and extensive in the gopher brain, and vascular lesions were found in the brain, lungs, liver, kidneys, and heart. Viral antigens and viral RNA were localized to blood vessels, neurons, lungs, kidneys, and spleens, and the viral genome was detected in the urine of infected gophers [36,37].

### 4.2. Ferret Model

Ferrets represent another well-characterized animal model for NiV pathogenesis. Experimental challenges via oral or IN routes with 5 × 10^2^–5 × 10^4^ TCID_50_ viral particles induce respiratory and neurological manifestations, featuring extensive multisystemic vasculitis that predominantly affects the upper (rhinitis/tonsillitis) and lower respiratory tracts. Viral replication kinetics mirror human pathology, with robust detection in pulmonary and cerebral tissues, accompanied by histopathological lesions consistent with clinical NiV infections [38]. This fidelity to human disease progression establishes ferrets as a physiologically relevant model for live-virus investigations.

### 4.3. African Green Monkey Model

The African green monkey model is currently the gold standard for NiV infection because NiV-infected monkeys exhibit vasculitis, endothelial syncytia, and acute respiratory and neurological symptoms leading to death, similar to the symptoms and course of human infections; this accurately generalizes human pathogenesis and is applicable to a variety of routes of attack, including aerosol transmission [39,40,41,42]. In a pathogenesis study, subjects receiving 2.5 × 10^3^–1.3 × 10^6^ PFU of NiV Malaysia strain via intratracheal and oral routes exhibited rapid viral dissemination to multiple organ systems within 72–96 h post-inoculation, progressing to terminal stages at 9–12 days post-challenge (dpc). Clinical manifestations paralleled human infection profiles: acute respiratory distress syndrome (ARDS), pyrexia, anorexia, lethargy, depressive behavior, and neurological deficits. Virological monitoring confirmed nasal viral shedding through RT-PCR detection in swab samples at 4–7 dpc [39]. Post-exposure blood samples were collected at 4, 7, 10, 14, 21, and 28 days, and at the endpoint of infection and death or detection (35 days), and were examined by the null spot test and quantitative PCR. The results show that no viraemia and NiV replication were detected in the vaccinated group, and that adaptive immune responses were correlated with vaccine-mediated protection by transcriptomics analysis [42]. While this model’s genetic proximity to humans and high-fidelity recapitulation of clinical manifestations establishes its biological supremacy, operational limitations persist. Stringent biosafety containment requirements (BSL-4), combined with substantial per-capita costs and complex husbandry needs, constrain its broad implementation, necessitating continued development of complementary model systems.

### 4.4. Cavia Porcellus Model

Cavia porcellus have been utilized as alternative models for NiV infection studies. Experimental evidence confirms viral presence in the urine, placental tissues, and amniotic fluid of infected animals through PCR detection [43,44,45]. However, IN inoculation with 6 × 10^5^ PFU of the NiV Malaysia strain failed to induce clinical manifestations, suggesting that this route of attack may not be conducive to the initiation of infection [35].

When Cavia porcellus were infected by IP with NiV at a dose of 5 × 10^4^ TCID_50_, mild clinical signs developed, including fur folds and mild ataxia 7–9 days after infection [46]. Another study in which the Malaysian strain of NiV was inoculated by IP showed that a dose of 6 × 10^4^ PFU caused severe clinical disease in Cavia porcellus, with most of them dying 4–8 days after inoculation [44]. Immunohistochemistry showed that histopathological lesions and high viral loads were detected in Cavia porcellus tissues such as lymph nodes, spleen, blood vessels, bladder, reproductive tract, and meninges 4–8 days after inoculation [44]. Pathological comparisons with human infections show partial overlap, notably attenuated pulmonary involvement but exacerbated urogenital damage characterized by mucosal ulcerations and submucosal vasculitis. The discrepancy in tissue tropism—particularly reduced lung pathology and enhanced urogenital manifestations—highlights species-specific differences in viral pathogenesis [44]. While the IP route effectively establishes systemic infection, current data indicate incomplete recapitulation of human respiratory disease patterns. Further validation is required to assess the model’s applicability for evaluating vaccine efficacy and therapeutic interventions, particularly regarding the standardization of challenge protocols and correlation between experimental outcomes and clinical relevance.

### 4.5. Cat Model

Experimental attacks on cats have shown that cats are highly sensitive to NiV. Studies have found that cats infected with NiV-M of 5 × 10^2.5^~5 × 10^3^ TCID_50_ through oronasal (ON) routes develop clinical diseases characterized by fever, depression, and respiratory distress. NiV can be detected in the lungs, spleen, bladder, and lymph nodes of cats with clinical symptoms [47,48]. Interestingly, the infectious viruses in these places are recyclable. During the outbreak of the epidemic, there were no records showing the transmission of NiV among cats. However, vertical transmission of NiV was observed in pregnant cats found during autopsies [49], and infectious viruses were detected in the placenta, fetus, and uterine fluid.

### 4.6. Pig Model

Pigs are intermediate hosts for NiV. NiV-infected pigs generally show mild clinical signs, including encephalitis and respiratory disease, while some individuals have no clinical signs and show self-limiting infection and low mortality. NiV can be detected in the nose and mouth of pigs with clinical signs [50]. Using pigs as an experimental model, it was found that subcutaneous or oral inoculation of NiV-Malaysia at doses up to 5 × 10^4^ TCID_50_ triggered respiratory and neurological syndromes in pigs, and the experimental pigs showed fever, lack of co-ordination, increased nasal mucous membrane secretion, and chronic coughing similar to those of naturally infected Malaysian pigs [48].

These NiV infections resulted in less severe morbidity than in humans. In addition, six 4-week-old piglets were orally inoculated with 2.5 × 10^5^ PFU of NiV. Four pigs showed mild clinical signs, one showed exudative dermatitis, and one showed neurological signs due to purulent meningitis, and the neutralizing antibody titer of the surviving animals was about 1280 after 16 days of virus attack [51]. To evaluate the immunoprotective effect of the NiV-F/G protein combination vaccine in pigs, crossbred female Long White pigs were used at 4 weeks of age and were domesticated for 1 week prior to the first experimental period, under BSL-4 conditions of animal husbandry. Vaccine-immunized and negative control groups were immunized from 5 to 7 weeks of age and challenged at 7 weeks of age with IN vaccination at a total dose of 2.5 × 10^5^ PFU. Sera were collected before vaccination and on days 7, 14, 21, and 28 after vaccination. Nasal lavage, pharyngeal swabs, and sera were collected before vaccination and 28 days after vaccination, and the samples were examined to determine the levels of neutralizing antibodies and cytokines, as well as the levels in brain and lung groups. The levels of neutralizing antibodies and cytokines in the samples were measured, as well as the extent of damage to brain and lung tissues, to assess the effectiveness of immunoprotection [22].

### 4.7. Mouse Model

Early challenge studies failed to induce clinical symptoms in mice with normal immune function. For example, one study found that Swiss mice inoculated with the NiV via nasal or intraperitoneal injection showed no clinical symptoms [24]. Another study using C57BL/6 mice, in which NiV–Malaysia strain (NiV-M) was administered orally, also did not cause illness. Only after direct intracerebral inoculation did the infected mice show symptoms such as weight loss, aggression, weakness, and paralysis. Later studies have utilized immunodeficient mice for genetic modification research to suppress the host’s antiviral immune response, achieving positive progress in the study of NiV mouse models [40]. It has been reported that type I interferon receptor knockout mice were administered a lethal dose of NiV through intraperitoneal injections, and infected mice developed neurological disease symptoms (such as head tilting and movement disorders) in the early stage, followed by paralysis in the later stage of infection [39]. There are also studies evaluating the susceptibility of transgenic mice lacking multiple innate pattern recognition signaling molecules to NiV infection. These proteins include myeloid differentiation factor 88 (MyD88) and TIR-domain containing adaptor inducing interferon-β (TRIF), mitochondrial antiviral signaling protein (MAVS), and stimulator of interferon genes (STING) [39,42]. Furthermore, it was found that MyD88/TRIF/MAVS/STING-KO mice developed symptoms of kyphosis, prone positioning, and paralysis, and died after being inoculated with the NiV-M strain via intraperitoneal injection [39]. Although the use of immunodeficient mouse models can achieve NiV infection and produce some clinical symptoms, the absence of an immune function affects the accuracy of antibody and vaccine efficacy evaluations.

These collective findings validate porcine models for vaccine efficacy testing despite species-specific attenuation of clinical severity. The model’s translational value resides in its capacity to mimic natural transmission dynamics while modelling the effect of immune protection through controlled challenge studies.

**Table 1 vaccines-13-00608-t001:** Summary of NiV infection animal models.

Species	Age	Virus Strain	Dose	Route	Clinical Symptoms	Applications	References
Golden Syrian hamster	2-month-old male	NiV-M	1~10^4^ pfu	IP	Tremor, limb paralysis	Antibody efficacy evaluation; pathogenesis study	[31,33]
Ferret	1~2 years old	NiV-M	5 × 10^2^~5 × 10^4^ TCID_50_	IN&ON	Multisystem vasculitis, respiratory distress, neurological signs (rhinitis, encephalitis)	Monoclonal antibody therapy assessment	[34,36]
African green monkey	Adult, 5–7 kg	NiV-M	2.5 × 10^3^~1.3 × 10^6^ pfu	IT&Oral	Acute respiratory distress, fever, neurological symptoms, multi-organ vasculitis	Gold standard for pathogenesis and vaccine durability studies	[40,41]
Cavia pocellus	4-month-old male	NiV-M	6 × 10^4^~1 × 10^7^ pfu	IP	Mild ataxia (IP); severe vasculitis and genital tract inflammation (high-dose IP)	Virus excretion research; preliminary vaccine evaluation	[44,46]
Cat	1~2 years old	NiV-M	5 × 10^2.5^~5 × 10^3^ TCID_50_	ON	Fever, increased respiratory rates, vomiting, decreased grooming, depression, dyspnea	Preliminary vaccine evaluation	[18]
Pig	6-week-old female	NiV-M	5 × 10^4^ TCID_50_	Oral&IH	Fever, respiratory distress, ataxia, exudative dermatitis, septic meningitis	Vaccine protection evaluation (e.g., F/G protein vaccine)	[18,47,49]
Mouse	4-month-old male	NiV-M	6 × 10^4^ pfu	IP	Vasculitis, lymphocytic meningitis and inflammation ofthe genitourinary tract	Susceptibility assessment; pathologic studies; pathogenesis studies; antibody efficacy evaluation	[52]

Note: IP, intraperitoneal inoculation; IN, intranasal inoculation; IT, thoracic inoculation; ON, oronasal; Oral, oral inoculation; IH, intradermal hypodermic.

## 5. Vaccine Research Progress

Although NiV has shown a pandemic trend in recent years, there is currently no effective treatment for NiV infection and no approved human NiV vaccine. The NiV vaccine was developed with two glycoproteins, F and G, as the main targets. Three vaccine candidates (HeV-sG-V, PHV02, and mRNA-1215) and a monoclonal antibody (m102.4) are currently in phase I clinical trials [53]. Other vaccines in development include subunit vaccines, virus-like particle vaccines, recombinant virus vaccines, inactivated vaccines, DNA vaccines, mRNA vaccines, and nanoparticle vaccines. Some vaccine candidates have been shown to provide complete protection in preclinical trials in hamsters, cats, ferrets, African green monkeys, and porcine models.

### 5.1. Subunit Vaccines

Subunit vaccines with well-defined target antigens are widely used in the vaccine field and have been developed to overcome the safety concerns of inactivated vaccines, offering advantages such as stability, ease of manufacture, and modifiable immune responses [54]. In 2012, a subunit vaccine expressing HeV soluble G proteins (HeV-sG-V) was licensed for the immunization of horses in Australia [34,55]. Experiments have shown that HeV-sG-V also protects cats, ferrets, African green monkeys, and horses against lethal NiV and HeV challenge. In the African green monkey model, a single dose of HeV-sG-V vaccine prevented HeV or NiV infection. A dose of 0.1 mg of HeV-sG-V protects ferrets against NiV-BD 7 days after vaccination. Two vaccinations provide at least 12 months of immune protection in ferrets [56]. The vaccine is currently in phase I clinical trials. In regions with severe outbreaks of NiV encephalitis, this vaccine could serve as an effective emergency vaccine to block the potential spread of NiV during outbreak situations [34].

### 5.2. Virus-like Particle Vaccines

Virus-like particles (VLPs) are potent immunogens that resemble viruses in size and structure. However, due to the absence of viral nucleic acids and polymerase activity, VLPs have a good safety profile [57]. Additionally, VLP proteins expressed by mammalian cells are capable of N-glycosylation and O-glycosylation modifications [58]. NiV M, G, and F proteins were expressed simultaneously in mammalian cells, resulting in the successful construction of virus-like particles (VLPs) that were fibrous and appeared identical to wild-type virions under electron microscopy. Mice were protected against the lethal effects of NiV after either a single immunization or three doses [58,59].

### 5.3. Recombinant Virus Vaccines

The main advantage of a recombinant viral vector vaccine is its rapid development and low side effects. It has shown strong immune responses in animal models. Many studies have been conducted on recombinant NiV viral vector vaccines, using viruses such as vesicular stomatitis virus (VSV), rabies virus (RABV), adeno-associated virus, chimpanzee adenovirus, canary pox virus, Newcastle disease virus, cowpox virus, and measles virus. For example, in an evaluation of the durability of NiV vaccine protection, African green monkeys were inoculated intramuscularly with a 1 × 10^7^ PFU dose of the rVSV-ΔG-NiV_BG_ vaccine or a non-specific rVSV vector control expressing the Ebola virus glycoprotein (rVSV-ΔG-EBOV-GP). One year after the vaccination, a challenge test was performed in which the green monkeys were intranasally inoculated with a 5 × 10^3^ PFU dose of the rVSV-ΔG-NiV_BG_ vaccine, which resulted in death. The lethal challenge with the NiV Bengal strain showed that the rVSV-ΔG-NiV_BG_-vaccinated monkeys exhibited no obvious clinical symptoms; only a few showed transient anorexia, and all survived until the study endpoint at 35 days post-infection. In contrast, the control group developed severe symptoms of infection by 7–9 days post-infection, meeting the criteria for euthanasia. Meanwhile, a replicable recombinant VSV vector vaccine (PHV02) encoding the NiV-MY G and Ebolavirus (EBOV) G genes protects ferrets and monkeys from lethal attacks by NiV after a single injection [19,60]. Originally developed by the National Institute of Allergy and Infectious Diseases (NIAID) in the United States, PHV02 has been further developed by Public Health Vaccines LLC (Cambridge, MA, USA). in collaboration with Crozet BioPharma and CEPI, and is currently in Phase I clinical trials [53]. The RABV vector is a well-established vaccine vector that lacks neurotoxicity and stably and efficiently expresses exogenous proteins [61]. Studies have shown that the intramuscular injection of C57BL/6 mice with either a single dose of live RABV vector vaccine expressing NiV-G protein (NIPARAB) or with two doses of vaccine inactivated with β-propanolactone resulted in both immunized groups showing specific conversion of serum IgG, with the inactivated vaccine group generating higher titers of specific antibodies [62]. The wide host range of adenoviruses and their low pathogenicity to humans make adenoviral vector systems widely suitable for protein expression. Immunization of hamsters with adenoviruses expressing NiV-G proteins provided complete protection against NiV infection and partial protection against HeV infection [63]. The chimpanzee adenoviral vector Ox1 is a replication-defective monkey adenoviral vector with the advantages of low reactogenicity, high immunogenicity, no vector replication after immunization, and high safety. A chimpanzee adenovirus vector vaccine encoding the NiV-G protein has been shown to protect hamsters against a lethal NiV challenge [64]. In addition, canarypox virus expressing NiV-F or G proteins (ALVAC-F/G) induced high levels of neutralizing antibodies when administered to pigs and followed by a booster vaccination 14 days after the initial vaccination [22]. Similarly, a cowpox virus-based vaccine expressing NiV glycoproteins (VV-NiV.F and VV-NiV.G) was found to protect hamsters against lethal NiV attacks; passive transfer of antibodies induced by both glycoproteins still protected the animals [65]. A measles virus vector vaccine expressing the NiV-G protein was tested twice in hamsters and African green monkeys and demonstrated complete protection against NiV [22].

### 5.4. Inactivated Vaccines

After NiV was concentrated and inactivated by binary ethylenimine (BEI), and purified by sucrose gradient centrifugation, the viral proteins, composed mainly of M and N proteins, were obtained, and BALB/c mice were immunized to prepare mouse monoclonal antibodies. These monoclonal antibodies can recognize the NiV antigen in ELISA, immunofluorescence, Western blot, and immunohistochemical detection [65]. Previous studies have shown that the use of BEI to inactivate paramyxovirus (such as Newcastle disease virus) is twice as effective and also retains the conformation of antigenic epitopes compared with formalin and betaprolactone [66,67].

### 5.5. DNA Vaccines

In a mouse model, the DNA immune response of NiV codon-optimized envelope glycoprotein genes F and G was studied. Compared with the F gene, G gene immunization caused a more pronounced specific serum IgG response and neutralizing antibody response, indicating that G gene DNA immunization is a potential vaccine strategy against NiV [68]. The advantages of this vaccine include high safety (no live virus component) and easy scale-up and ambient storage, but requires enhanced delivery by electroporation or adjuvant to increase immunogenicity [13]. Vaccine candidates have been shown to be effective in neutralizing the virus and improving survival in animal models (e.g., hamsters, primates), but have not yet entered the clinical stage, and their long-term protection and population applicability still need to be further evaluated.

### 5.6. mRNA Vaccine

The mRNA vaccine platform has been widely pursued, particularly in the recent past, as it has the potential to allow for rapid vaccine development against emerging pathogens. RNA vaccines for NiV are delivered by optimizing mRNA sequences encoding key antigens of the virus (e.g., fusion protein F or attachment glycoprotein G) encapsulated in lipid nanoparticles (LNPs), which are then inoculated with a host cell that uptakes the mRNA and expresses the targeted protein. After inoculation, the host cell takes up the mRNA and expresses the target protein, which in turn activates humoral immunity (production of neutralizing antibodies) and cellular immunity (induction of T-cell responses), thus providing protection [69]. The advantages of this vaccine include a short development cycle, high safety (no risk of genomic integration), rapid response to viral mutations, and an LNP delivery system that effectively enhances mRNA stability and transfection efficiency [70]. Preclinical studies have shown that mRNA vaccines based on NiV F or G proteins can induce high levels of neutralizing antibodies and significantly improve survival in animal models (e.g., hamsters, nonhuman primates), but their long-term immune durability, large-scale production processes, and effects on population vaccination still need to be further validated [53].

### 5.7. Nanoparticle Vaccine

In recent years, the field of vaccine research and development has continued to expand its boundaries. Not only are conventional vaccines making steady progress, but the field of NiV nano-vaccines is also advancing through in-depth research and continuous innovation. Nanotechnology-based NiV vaccines have been widely used in the development of vaccines against various infectious pathogens and tumors due to their unique nanoparticle structure, which offers strong advantages in specific antigen delivery, multivalent binding of B cell receptors, immune activation, and other key aspects [14]. Ferritin, a natural nanoscale protein cage with good biocompatibility and self-assembly properties, is often used as a carrier for NiV nanovaccines [14]. Researchers designed a ferritin-based self-assembling nanoparticle (NiV G-ferritin) displaying NiV G head domains on the surface and evaluated the immune response induced by soluble NiV G head domains (NiV sG) or NiV G-ferritin. The results show that both vaccines protected hamsters against lethal NiV attacks. NiV G-ferritin induced significantly faster, broader, and higher seroneutralization responses against three pathogenic Henipaviruses (NiV-Malaysia, NiV-Bangladesh, and Hendra virus) compared to NiV sG. In addition, NiV G-ferritin induced durable neutralizing immunity in mice, as antisera potently inhibited NiV infection even 6 months after the third immunization [14]. Other nanoparticle vaccines, FeNP-sG and FeNP-Ghead, display the soluble G protein (sG) and G protein head structural domain (G_head_) of NiV, respectively. Antigen binding to FeNPs using SpyCatcher/SpyTag technology significantly enhanced their immunogenicity. In a mouse model, FeNP-sG and FeNP-G_head_ induced significant neutralizing antibody levels and T-cell responses, with the immune response to FeNP-sG being superior to that of FeNP-Ghead. In a hamster model, two doses of 5 μg FeNP-sG provided 100% protection against lethal attack by NiV [71].

## 6. Treatment

Most therapeutic measures for NiV encephalitis are supportive and include anticonvulsant drugs, treatment of secondary infections, mechanical ventilation, and rehabilitation. In addition, several antibodies and drugs, such as the monoclonal antibody drug m102.4, favipiravir (T-705), ribavirin, and acyclovir have been considered. Among these, m102.4 is currently in Phase I clinical trials [52].

### 6.1. Monoclonal Antibody

Monoclonal antibodies play an important role in biopharmaceuticals because of their high antigenic affinity, specificity, and long half-life. The monoclonal antibody m102.4, which targets HeV surface glycoproteins, has shown pre- and post-exposure prophylaxis efficacy against HeV and NiV in animal studies. It targets the ephrin B2/B3 receptor-binding region of the HeV-G and NiV-G proteins and effectively cross-reacts with both viruses. It has been shown to protect ferrets against lethal NiV infection [72]. In an animal study involving 14 African green monkeys, none of the 12 monkeys treated with m102.4 developed an infection or died after being injected with a lethal dose of NiV, whereas the 2 untreated monkeys developed severe infections that progressed to encephalitis and acute respiratory distress syndrome [72]. Currently, m102.4 is in phase I clinical trials for the treatment of NiV encephalitis [73,74].

In therapeutic evaluations, juvenile Syrian hamsters (3–5 weeks old) were intranasally inoculated with 5 × 10^6^ PFU of NiV Bangladesh strain followed by intraperitoneal administration of 10 mg/kg monoclonal antibodies (HENV-117/HENV-103) at 24 h post-infection (*n* = 5/group). Longitudinal monitoring over 28 days showed that both types of antibodies reduced morbidity and mortality, whereas untreated controls developed acute respiratory lesions that progressed to fatal encephalopathy [35]. This model partially recapitulates human disease manifestations, including pulmonary necrosis and meningeal inflammation. However, systematic investigations are required to refine characterization of infection route-dependent pathogenesis, temporal symptom progression, and neuroinflammatory mechanisms.

In therapeutic protocols, female ferrets (0.75–1 kg body weight) were challenged intranasally with 5 × 10^3^ PFU of NiV Malaysia strain under ketamine-promazine-xylazine anesthesia. Treatment cohorts received IP monoclonal antibodies (h5B3.1/HENV-26; 15–20 mg/kg) at 3- and 5 dpi, while controls received a placebo [75,76]. Monitoring included daily clinical scoring, thermometry, gravimetric analysis, and serial blood sampling on days 0, 3, 6, 8, 11, 21, and 34 dpi. Terminal necropsy at 34 dpi involved comprehensive tissue collection (lung, liver, spleen, kidney, adrenal gland, pancreas, brain) for viral load quantification via a quantitative real-time polymerase chain reaction (qRT-PCR) assay. Both groups of antibody-treated ferrets survived to the end of the study, gaining weight and showing mild clinical signs mediated by NiV infection, with no significant pathological changes observed at autopsy at the end of the study. Postmortem analyses revealed no significant histopathological abnormalities in therapeutic groups, confirming antibody-mediated protection [75]. The model’s translational value stems from its capacity to recapitulate critical aspects of human NiV infection, particularly the temporal progression of respiratory-to-neurological disease transition. This pathophysiological congruence, combined with a reproducible response to antibody therapies, solidifies ferrets as a preferred model for preclinical evaluation of antiviral countermeasures.

### 6.2. Favipiravir (T-705)

Favipiravir (T-705), an antiviral drug manufactured by Fuji in Japan, is an inhibitor of the RNA-dependent RNA polymerase of NiV. It not only inhibits the replication of the viral genome and the release of progeny virus in infected cells, but also prevents the spread of the virus to uninfected cells. The drug protected hamsters from a lethal NiV attack in a Syrian hamster model when administered orally twice a day or subcutaneously once a day, with the challenge occurring after 14 days [77].

### 6.3. Ribavirin (Virazole)

Following the identification of NiV, outbreaks in Malaysia and Singapore were treated empirically with ribavirin [78,79]. Ribavirin is known for its broad-spectrum anti-DNA and RNA viral activity and its ability to cross the blood–brain barrier [80]. A 36% reduction in mortality with ribavirin was reported in an open-label study of 140 patients and 54 controls, but its efficacy has not been confirmed in subsequent studies using animal models [80,81].

## 7. Prevention and Control

Due to the limited treatment options for NiV infection, prevention should be emphasized. In terms of animal prevention, appropriate measures can be implemented. In areas where NiV poses a threat, fruits should be washed before consumption, and steps should be taken to prevent farm animals from becoming infected by consuming fruits contaminated by bats. Farms should appropriately reduce stocking density to avoid the spread of diseases among animals, and the animals should be kept as far away as possible from fruit trees that attract bats. In terms of human prevention, travel to countries where NiV is endemic should be avoided, and surveillance systems should be strengthened to prevent the recurrence and cross-border spread of NiV. For individuals working in fields or farms infected with NiV, personal protective measures such as masks, goggles, gloves, gowns, and boots are recommended. Additionally, frequent handwashing and the use of disinfection equipment should be encouraged.

NiV is unstable in vitro and sensitive to heat. It can survive for 1 h at 70 °C and can be completely inactivated by heating to 56 °C for 30 min or 100 °C for 15 min. NiV, like other paramyxoviruses, can be easily inactivated by common disinfectants such as 0.5% sodium hypochlorite, 0.5% phenol, or 75% ethanol, as well as by soaps and other cleaning agents [82].

## 8. Discussion

NiV, due to its high lethality and potential pandemic risk, has become a critical concern in global public health. The development of NiV vaccines is of paramount importance for viral prevention. As a proactive strategy, NiV vaccine research can help curb cross-regional transmission and prevent scenarios similar to the large-scale outbreaks seen with SARS-CoV-2. By utilizing vaccines for early-stage containment, the spread of the virus can be limited, reducing the strain on healthcare systems and ensuring socioeconomic stability. At the animal level, veterinary vaccines play a crucial role in preventing cross-species transmission. NiV circulates among animal reservoirs before spilling over into humans. Widespread vaccination of animal hosts to reduce viral prevalence in these populations would help mitigate zoonotic spillover risks at the source, aligning with the One Health concept that promotes synergistic advancements in both human and animal health.

Despite recent progress in NiV vaccine research, significant challenges remain. First, no approved human vaccine exists, necessitating further research and validation to bridge the gap between preclinical studies and human clinical trials. Second, the immunogenicity and protective efficacy of candidate vaccines must be optimized to enhance their potency and durability. Advances in biotechnology are driving the development of novel strategies. In vaccine design, rational approaches based on structural biology and bioinformatics are becoming increasingly important. By resolving the three-dimensional structures of NiV antigens, researchers can elucidate antigen–antibody interaction mechanisms, facilitating the design of vaccines with enhanced immunogenicity and specificity. Furthermore, innovative delivery technologies, such as nanotechnology and lipid-based systems, show promise for improving the stability and delivery efficiency of nucleic acid vaccines, thereby accelerating their clinical translation. However, the extreme lethality of NiV, stringent development requirements, and sporadic outbreak patterns complicate the execution of large-scale Phase III clinical trials. Additionally, limited funding from pharmaceutical companies for trials in low- and middle-income countries—where NiV outbreaks predominantly occur but resources are scarce—remains a significant barrier to clinical development.

In animal model research, future efforts should prioritize precision, efficiency, and relevance to human infection dynamics. On one hand, genome-editing technologies can be leveraged to develop humanized animal models (e.g., humanized mice) that more accurately recapitulate human immune responses and pathological processes. On the other hand, integrating multiple animal models with multi-omics technologies will enable comprehensive investigations into NiV pathogenesis, accelerating the development of vaccines and therapeutics.

To address these challenges, the global scientific community must adopt an open, collaborative approach. By consolidating multidisciplinary resources spanning biology, medicine, pharmacology, and materials science, a cross-disciplinary innovation framework can be established to accelerate the translation of vaccines from bench to bedside. Only through the deep integration of diverse disciplines and robust international cooperation can we fully unravel NiV pathogenesis, develop safe and effective vaccines, and achieve sustainable control of NiV. This will not only safeguard global public health but also promote the harmonious coexistence of human, animal, and environmental health, ensuring the stability and balance of ecosystems.

## 9. Conclusions

This article systematically reviews the current development status of NiV-infected animal models and their applicability in evaluating protective effects. The live-virus-infected animal models of NiV, such as golden hamsters, ferrets, cats, pigs, African green monkeys, and Cavia porcellus, can simulate human clinical infection characteristics and serve as critical tools for elucidating pathogenic mechanisms and assessing vaccine/antibody efficacy. Among these, golden hamsters, ferrets, pigs, and African green monkeys have been utilized for evaluating vaccines and antibodies, providing essential insights for preventive and therapeutic strategies. However, these live-virus models face limitations: reliance on laborious virus extraction methods from diverse tissues, complex experimental workflows, and stringent biosafety requirements due to NiV’s high virulence and lethality, which collectively increase costs and restrict research scalability. In contrast, pseudovirus-based models offer advantages such as lower biosafety risks (BSL-2), cost-effectiveness, and real-time infection monitoring, making them promising for high-throughput drug screening. However, their consistency with live-virus models in mimicking infection dynamics and immune responses remains unvalidated, and standardized evaluation criteria are lacking. Future efforts should focus on harmonizing validation protocols between both models to assess pseudovirus suitability as a surrogate for live-virus “gold standard” evaluations. Advancements in pseudovirus vector engineering and multimodal evaluation systems are expected to enhance their utility in accelerating NiV countermeasure developments, ultimately supporting safer and more efficient translational research.

## Figures and Tables

**Figure 1 vaccines-13-00608-f001:**
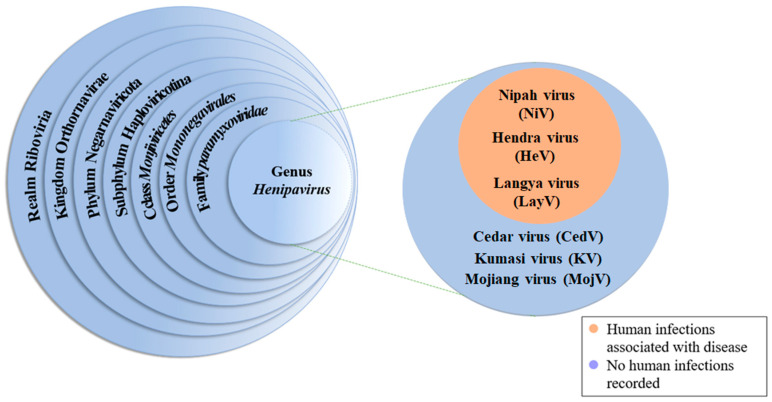
Species classification of NiV.

**Figure 2 vaccines-13-00608-f002:**
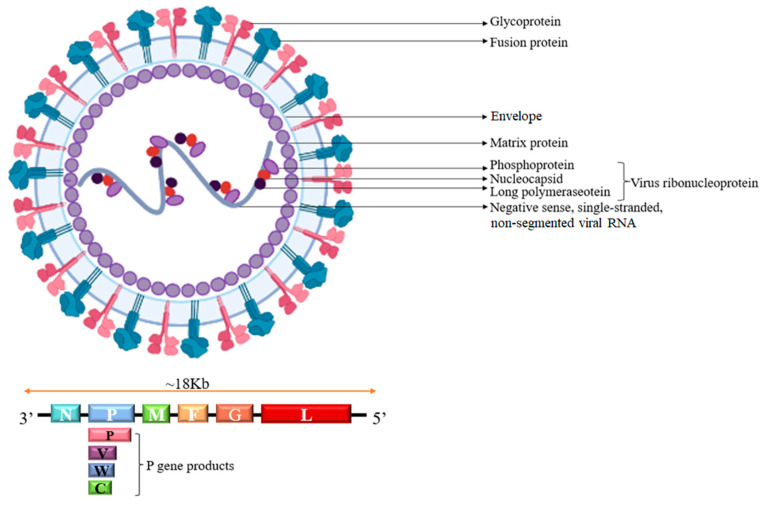
Structure and encoded proteins of NiV.

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
