# Peer review of "Vaccines and Animal Models of Nipah Virus: Current Situation and Future Prospects"

_vaccines, 2025, doi:10.3390/vaccines13060608_

Round 1
Reviewer 1 Report
Comments and Suggestions for Authors
The authors provide an interesting overview of NiV studies in different animals and also discuss various vaccines and monoclonal antibody-based treatment strategies.
Comments are listed below:
Please add treatments for NiV infection in humans to the introduction, including monoclonal antibody therapy. Please move all studies using monoclonal antibodies from Section 4 to Section 6.
Some of the cases described were infected with live virus, while others used vaccines, such as the rVSV-ΔG-NiVBG vaccine (section 4.3). Please add a section describing vaccines and separate viruses from vaccines.
Lines 31-34. Please add more details about the symptoms of the disease? Have serious complications and deaths been reported? Please describe the histopathological lesions consistent with clinical NiV infection. Give references.
Line 77. " the F and G proteins" are mentioned. Please move this after the characterisation of these proteins.
Lines 112-118. Please provide references for these studies.
Table 1. There is no reference to this table in the text of the manuscript. Please summarise all studies in this table.
Paragraph 5. Please provide the drug IDs of the clinical trials in the ClinicalTrials.gov database (https://clinicaltrials.gov/). Also, monoclonal antibodies are not vaccines, so they should be listed separately from vaccines.
Paragraph 7. Please add a description of the major modes of transmission of NiV.
Author Response
Response to Reviewer Comments
Dear Editor and reviewer:
Dear Editor/Reviewer.
Thank you very much for your review and comments on our paper, entitled " Vaccines and Animal Models of Nipah Virus: Current Situation and Future Prospects " and your time regarding for our revision. We also appreciate all the critical comments from you and reviewers. Yours professional opinions have greatly improved the quality and professionalism of manuscripts. Here, we deeply appreciate the comments and suggestions and have carefully considered each one. The following is a point-by-point response to all those comments and a list of changes we have made to the manuscript (The manuscript corresponds to sections with yellow background).
We have carefully words and sentences with high repetition rates, as well as revised the manuscript accordingly. With these improvements, we hope that the current version can meet the Journal’s standards for publication. Beyond that, if you still have any suggestions or questions about our revisions, please feel free to let us know and we will be more than happy to collaborate further.
Thank you again for your time and help.
Sincerely yours!
Chaoxiang Lv, Ph.D
Professor and director
Sichuan, Luzhou, China.
E-Mail: lvchaoxiang@126.com
Point-by-point responses to the comments of the Editor and reviewers, and a list of changes are:
(The comments of the Editor and reviewers are in italics and blue color, which are followed by our responses.)
Reviewer(s)' Comments to Author:
Reviewer: 1
Comments and Suggestions for Authors
The authors provide an interesting overview of NiV studies in different animals and also discuss various vaccines and monoclonal antibody-based treatment strategies.
Major points:
Comment 1. Please add treatments for NiV infection in humans to the introduction, including monoclonal antibody therapy. Please move all studies using monoclonal antibodies from Section 4 to Section 6.
Response 1: Dear reviewer, thank you very much for your recognition and attention to our study. We appreciate your suggestions for enhancing the clinical relevance of the introduction. We have information on treatment for NiV infection in humans to the introduction, detailing the current treatment strategies for human NiV infection, including monoclonal antibody therapy and supportive treatment. All the studies originally related to monoclonal antibodies in Section 4 have been reclassified into Section 6 "Treatment" and discussed in combination with other human treatment methods. This adjustment optimizes the logical structure by distinguishing between preventive (vaccine) and therapeutic (monoclonal antibody) strategies. Thank you sincerely again for your careful comments on the manuscript.
Comment 2. Some of the cases described were infected with live virus, while others used vaccines, such as the rVSV-ΔG-NiVBG vaccine (section 4.3). Please add a section describing vaccines and separate viruses from vaccines.
Response 2: We would like to thank this respected reviewer for his useful comments. We recognize the importance of clearly distinguishing between live virus research and vaccine research. Section 5 "Vaccine research progress" has now been newly added, systematically summarizing preclinical and clinical vaccine research. The virus infection model is retained separately in Section 4 "Animal Models", thereby clearly distinguishing the model system from the intervention strategy.
Comment 3. "the F and G proteins" are mentioned. Please move this after the characterisation of these proteins.
Response 3: We would like to thank this respected reviewer for his useful comments. The discussion on F protein and G protein has been moved to the Section 3 "The structure of NiV". This adjustment ensures logical fluency. The protein characteristics are introduced first, and then its application in vaccine design is discussed. Thank you sincerely again for your careful comments on the manuscript.
Comment 4. Lines 112-118. Please provide references for these studies.
Response 4: Thank you very much for your patience in reviewing this manuscript and for your valuable suggestions, which will make our article more scientifically sound. We have added references to the research mentioned in lines 116-118 (lines 121-124 in the revised manuscript)。Amino acid similarity between NiV and HeV ranges from 68% to 92%, with 88% and 83% similarity between the F and G proteins of the two, respectively (PMID: 38912755). Thus, NiV and HeV exhibit significant or complete cross-reactivity (PMID: 33558494). All claims have now been supported by the original literature to enhance the credibility of the manuscript.
Comment 5. Table 1. There is no reference to this table in the text of the manuscript. Please summaries all studies in this table.
Response 5: We would like to thank this respected reviewer for his useful comments. Table 1 has been cited in "Animal Models" of the manuscript (line 130), which now contains a comprehensive summary of the animal model and its application in NiV studies. The table legend has been expanded to clarify its purpose and key parameters (such as genetic background, clinical symptoms, and applications).Thank you sincerely again for your careful comments on the manuscript.
Comment 6. Paragraph 5. Please provide the drug IDs of the clinical trials in the ClinicalTrials.gov database (https://clinicaltrials.gov/). Also, monoclonal antibodies are not vaccines, so they should be listed separately from vaccines.
Response 6: We would like to thank this respected reviewer for his useful comments. We added the corresponding drug numbers, such as Favipiravir (T-705) and Ribavirin (Virazole), in the "Treatment" section of the manuscript. Additionally, monoclonal antibody therapy is now listed separately in the sub-section of Section 6.1, separated from vaccine candidates, to emphasize its role as a therapeutic (rather than preventive) intervention. Thank you sincerely again for your careful comments on the manuscript.
Comment 7: Paragraph 7. Please add a description of the major modes of transmission of NiV.
Response 7: Thanks so much for your careful work. We describe in the introduction the main routes of transmission of NiV, including zoonotic transmission from fruit bats (storage hosts) to humans through contact with infected animals or contaminated food, and human-to-human transmission through respiratory droplets or contact with body fluids. “NiV infection is mainly transmitted through direct contact with respiratory droplets, nasal secretions and tissues of infected animals. It can also be spread by consuming food contaminated with the urine, feces or saliva of infected bats. It has been reported that NiV infection can be transmitted from person to person, mainly through close contact with the secretions and excretions of infected patients(4). After humans are infected with NiV, they may show symptoms such as fever, headache, dizziness, vomiting, decreased consciousness, segmental myoclonus, reduced reflexes, hypotonia, and abnormal head-eye reflexes, with a case-fatality rate as high as 75%.”
Again, thanks for the reviewer’s concern and support for our research. We believe that our research will be more perfect and reliable through the above improvement measures. We look forward to receiving your further guidance and recognition.

Reviewer 2 Report
Comments and Suggestions for Authors
In this manuscript, Lv and colleagues review animal models, vaccination approaches, and antiviral strategies for Nipah virus. The review does a good job of summarizing the large variety of animal models that have been established, and this will be helpful to readers in the field. In the second half of the review, authors describe a number of vaccine and antiviral studies that illustrate productive use of these animal models.
I did find a number of mistakes during my review of the manuscript, and I have some concerns regarding accuracy in general. In addition many of the discussions are quite superficial and should be presented in a more thorough and systematic way. Specific concerns:
1. Authors are inconsistent in the depth of discussion provided for the various vaccination strategies. For example, discussions of mRNA vaccines and adenovirus vectored vaccines are quite superficial, even though a wealth of context exists in the literature. The section on DNA vaccines is two sentences long and does not appropriately introduce the concepts needed for readers to fully understand the technology. The writing in Section 5.4 is unclear.
2. The logic behind the organization of Section 5 is unclear. It would make sense to discuss the vaccine candidates that have advanced to clinical trial first, and then the more experimental approaches.
3. Line 74. Coomassie virus??
4. Line 81. Nucleotide similarities??
5. Fig. 1. Why is HeV in both categories (human infections, and no human infections recorded)?
6. Line 87 "Like all other paramyxoviruses, the NiV genome can be divided into six parts, as each N must bind six nucleotides". As written, I think this means something different from what the authors intend.
7. Line 95 "The accessory protein genes (V/W/C) are encoded by the P gene, which share the same N-terminal structural domain but have different C-terminal structural domains due to the discontinuity of the RNA polymerase during mRNA transcription." I don’t think this is how it works for C protein.
8. Line 118. I think hamsters and gophers are not the same kind of animal.
Comments on the Quality of English LanguageAn additional round of proofreading would be beneficial.
Author Response
Response to Reviewer Comments
Dear Editor and reviewer:
Dear Editor/Reviewer.
Thank you very much for your review and comments on our paper, entitled " Vaccines and Animal Models of Nipah Virus: Current Situation and Future Prospects " and your time regarding for our revision. We also appreciate all the critical comments from you and reviewers. Yours professional opinions have greatly improved the quality and professionalism of manuscripts. Here, we deeply appreciate the comments and suggestions and have carefully considered each one. The following is a point-by-point response to all those comments and a list of changes we have made to the manuscript (The manuscript corresponds to sections with yellow background).
We have carefully words and sentences with high repetition rates, as well as revised the manuscript accordingly. With these improvements, we hope that the current version can meet the Journal’s standards for publication. Beyond that, if you still have any suggestions or questions about our revisions, please feel free to let us know and we will be more than happy to collaborate further.
Thank you again for your time and help.
Sincerely yours!
Chaoxiang Lv, Ph.D
Professor and director
Sichuan, Luzhou, China.
E-Mail: lvchaoxiang@126.com
Point-by-point responses to the comments of the Editor and reviewers, and a list of changes are:
(The comments of the Editor and reviewers are in italics and blue color, which are followed by our responses.)
Reviewer(s)' Comments to Author:
Reviewer: 2
Comments and Suggestions for Authors
In this manuscript, Lv and colleagues review animal models, vaccination approaches, and antiviral strategies for Nipah virus. The review does a good job of summarizing the large variety of animal models that have been established, and this will be helpful to readers in the field. In the second half of the review, authors describe a number of vaccine and antiviral studies that illustrate productive use of these animal models.
I did find a number of mistakes during my review of the manuscript, and I have some concerns regarding accuracy in general. In addition many of the discussions are quite superficial and should be presented in a more thorough and systematic way. Specific concerns:
Major points:
Comment 1: Authors are inconsistent in the depth of discussion provided for the various vaccination strategies. For example, discussions of mRNA vaccines and adenovirus vectored vaccines are quite superficial, even though a wealth of context exists in the literature. The section on DNA vaccines is two sentences long and does not appropriately introduce the concepts needed for readers to fully understand the technology. The writing in Section 5.4 is unclear.
Response 1: Thanks so much for your careful work. The DNA vaccine and mRNA vaccine sections have been expanded and mechanism details have been incorporated in lines 77 to 116 of the manuscript. In the DNA vaccine section, we have added the advantages (high safety, easy storage) and disadvantages (requiring the addition of adjuvants) of this vaccine. Additionally, vaccine candidates have been shown to be effective in neutralizing the virus and improving survival in animal models (e.g., hamsters, primates), but have not yet entered the clinical stage, and their long-term protection and population applicability still need to be further evaluated. In the RNA vaccine section, in addition to comparing the advantages and disadvantages of this vaccine, we have also added some characteristics of the NiV RNA vaccine and the features of clinical research.
Section 5.4 has been rephrased to enhance clarity. We change the original text to "After NiV was concentrated and inactivated by binary ethylenimine (BEI), and purified by sucrose gradient centrifugation, the viral proteins, composed mainly of M and N proteins, were obtained, and BALB/c mice were immunized to prepare mouse monoclonal antibodies. These monoclonal antibodies can recognize the NiV antigen in ELISA, immunofluorescence, Western blot, and immunohistochemical detection. The previous studies have shown that the use of BEI to inactivate paramyxovirus (such as Newcastle disease virus) is twice as effective and and also retains the conformation of antigenic epitopes compared with formalin and betaprolactone.".
Again, thank you for your useful comments.
Comment 2: The logic behind the organization of Section 5 is unclear. It would make sense to discuss the vaccine candidates that have advanced to clinical trial first, and then the more experimental approaches.
Response 2: Thanks so much for your careful work. We sincerely appreciate the opportunity to clarify our description of the vaccine section in the manuscript. In the NiV vaccine section, we introduced different vaccines, aiming not only to echo the observations of the epidemic but also to provide an in-depth understanding of them. Considering the development path of vaccines, when describing vaccines, we first introduced Subunit vaccines, virus-like particle vaccines, Recombinant virus vaccines, and Inactivated vaccines. Secondly, from the perspective of biological macromolecules, we introduced DNA vaccines and mRNA vaccines respectively. Finally, we introduced the Nanoparticle vaccine. Thank you again for your advice!
Comment 3: Line 74. Coomassie virus??
Response 3: We would like to thank this respected reviewer for his useful comments. We accepted it and changed it in manuscript.
Comment 4: Line 81. Nucleotide similarities??
Response 4: Thanks so much for your careful work. The original text is “Genomic sequencing have identified two strains of NiV: NiV-B and NiV-M. Genetic characterization of the two strains demonstrated that NiV-B has a genome size of 18,252 nucleotides which is longer than NiV-M by six nucleotides. Even though both the strains share 91.8% similarity in nucleotide homology, NiV-B is considered to have higher fatality rates ”. The source is “Nipah Virus: Past Outbreaks and Future Containment” (PMID: 32325930). Furthermore, in the manuscript, we also cite literature to clarify this background. Thank you sincerely again for your careful comments on the manuscript.
Comment 5: Fig. 1. Why is HeV in both categories (human infections, and no human infections recorded)?
Response 5: Thanks so much for your careful work. HeV is now only listed in the "Human Infection" category (known to cause human diseases) and has been removed from the "No Record of Human Infection".Thank you sincerely again for your careful comments on the manuscript.
Comment 6: Line 87 "Like all other paramyxoviruses, the NiV genome can be divided into six parts, as each N must bind six nucleotides". As written, I think this means something different from what the authors intend.
Response 6: Thanks so much for your careful work. We have changed it to "Like all other paramyxoviruses, the NiV genome contains six genes corresponding to the nucleocapsid (N), phosphoprotein (P), matrix protein (M), fusion protein (F), glycoprotein (G), and large protein (L)."
Comment 7: Line 95 "The accessory protein genes (V/W/C) are encoded by the P gene, which share the same N-terminal structural domain but have different C-terminal structural domains due to the discontinuity of the RNA polymerase during mRNA transcription." I don’t think this is how it works for C protein.
Response 7: Thanks so much for your careful work. We have changed it to “The accessory proteins V/W/C are encoded through the variable open reading frame and RNA editing of the P gene, where the C protein is translated by the downstream start codon of the P mRNA”. This accurately reflects the molecular mechanism of C protein synthesis. Thank you sincerely again for your careful comments on the manuscript.
Comment 8: Line 118. I think hamsters and gophers are not the same kind of animal.
Response 8: Thanks so much for your careful work. It has been revised as hamster (such as Golden Syrian hamster, Table 1).
Again, thanks for the reviewer’s concern and support for our research. We believe that our research will be more perfect and reliable through the above improvement measures. We look forward to receiving your further guidance and recognition.

Round 2
Reviewer 1 Report
Comments and Suggestions for Authors
The new additions to the manuscript made a big difference. The quality of the paper had improved, and all my questions were addressed. No more comments.